# A Phase Model of the Bio-Inspired NbOx Local Active Memristor under Weak Coupling Conditions

**DOI:** 10.3390/mi15030390

**Published:** 2024-03-13

**Authors:** Xuetiao Ma, Yiran Shen

**Affiliations:** School of Electronics and Information Engineering, Hangzhou Dianzi University, Hangzhou 310018, China; mxt@hdu.edu.cn

**Keywords:** local active memristor, Chua’s unfolding principle, phase model, coupling function, interaction function, Fourier expansion

## Abstract

For some so-called computationally difficult problems, using the method of Boolean logic is fundamentally inefficient. For example, the vertex coloring problem looks very simple, but the number of possible solutions increases sharply with the increase of graph vertices. This is the difficulty of the problem. This complexity has been widely studied because of its wide applications in the fields of data science, life science, social science, and engineering technology. Consequently, it has inspired the use of alternative and more effective non-Boolean methods for obtaining solutions to similar problems. In this paper, we explore the research on a new generation of computers that use local active memristors coupling. First, we study the dynamics of the memristor coupling network. Then, the simplified system phase model is obtained. This research not only clarifies a physics-based calculation method but also provides a foundation for the construction of customized analog computers to effectively solve NP-hard problems.

## 1. Introduction

Chua was the first to propose the concept of a memristor [1,2], and it was not until May and June 2008 that *Nature* published three consecutive articles, reporting the discovery of memristors [3,4,5]. HP Laboratories discovered that a two-layer titanium dioxide film, sandwiched between two platinum sheets, exhibited the characteristics of a memristor. This finding was the first to theoretically and experimentally confirm the physical existence of nano memristors, causing great shock in both the industry and academia. Thus, memristors have become a new hot research field [6]. Memristors are a type of memory-based nonlinear resistor with nanoscale dimensions, precisely adjustable resistance, nonvolatility, and low power consumption characteristics. Their voltage–current pinched hysteresis characteristics and frequency dependence on periodic excitation signals are the main distinguishing features of memristors [7]. Current research indicates that memristors have extremely important application potential in fields such as non-volatile memory, artificial neural networks, logic circuits, and nonlinear circuits.

The direct application of a binary nonvolatile memristor is to build a new generation of nonvolatile resistive memory (ReRAM), which has the advantages of scalability, low power consumption, high density, and compatibility with CMOS. It is the preferred alternative for future Flash, SRAM, and DRAM. Research indicates that, in the future, ReRAM can be scaled down to below 10 nm, and the development of a 20 nm 1 Gb 2-layer 3D ReRAM has already been achieved. The density of 3D ReRAM can exceed that of 2D and 3D flash memory [8]. The cross-array of 8 nm × 8 nm memristors has been reported [9], and the development of 1 nm-level memristors is anticipated [10].

The exponential growth in data volume and computing demand has led to the Von Neumann bottleneck [11] in traditional storage computing architectures, and computing technologies driven by transistor density are gradually reaching their physical limits. The emergence of neural morphology computing, which uses neural morphology devices to simulate the behavior of neurons and synapses in the human brain to process information, has become the best candidate for the new generation of computing architecture due to its high parallelism, extremely low power consumption, and integrated storage and computing advantages. The nonvolatile and adjustable resistance characteristics of memristors can emulate the memory and weight regulation behavior of synapses, while local active memristors can emulate various firing characteristics of neurons [7,8,9,10], and the nanostructure of memristors can make neural networks highly integrated. Therefore, memristors can directly emulate the behavior of neurons and synapses from a physical level [12,13], naturally achieving a computing architecture that integrates memory and computing, which precisely meets the needs of neural morphology devices and has become one of the best candidates for current neural morphology devices.

Binary memristors have “0” and “1” logical states, which can be used to implement logical operations. Memristive logic circuits are the most promising alternative computing solution for traditional integrated circuits. Using memristors for both memory and processing functions, can break the computational framework of von Neumann’s separation of memory and computation, thus achieving a new architecture of integrated memory and computation. This is a more optimized approach for the new generation of computing machines. Logic circuits based on memristors mainly include memristor/CMOS hybrid logic circuits, memristor logic circuits, logic operation circuits based on programmable nanowire technology, embedded logic circuits, etc. [10,11].

In addition to the above applications, memristors can also be applied to amplifiers, filters, and nonlinear oscillation and chaos circuits [14]. Memristors provide another new development space for circuit design, especially in the field of memristive chaotic oscillators. Special phenomena, such as infinite equilibrium points (or line, plane, or even three-dimensional space equilibrium points), coexisting chaotic attractors (or parameter-free bifurcations), hidden chaotic attractors, and excessive stability, have been discovered [15,16,17]. Memristor chaotic circuits are not only easy to integrate, but also play an extremely important role in applications within chaotic neural networks. There is evidence to suggest that neural networks working under chaotic edge mechanisms can optimize neural computation and global search [18].

Recently, *Nature* published a third-order nano integrated circuit component based on a local active memristor [19]. This component can simulate 15 functions of a single neuron, and can exhibit complex dynamic characteristics of neurons, such as chaotic oscillation, unimodal periodic oscillation, and action potential under different DC biases.

After the discovery of memristors, Chua et al. expanded their research to include the concepts of memcapacitors and inductors [20] and conducted some fundamental theoretical studies. Although some phenomena with characteristics of memory containers and sensors have been preliminarily discovered, such as bistable elastic thin films [21] exhibiting chaotic characteristics under certain excitations, practical and useful artificial physical devices have not yet been realized.

Although Boolean logic has always been the backbone of digital information processing, there are some so-called NP-hard problems. The logical method is fundamentally inefficient [22,23], which inspires the use of alternative, more effective non-Boolean methods to obtain a solution to this problem [24,25,26,27,28]. Based on phase dynamics, it is a good idea to build an Ising computer to solve NP-hard problems. At present, there are many ways to implement the oscillator element [29,30,31,32,33,34,35,36,37,38,39,40]. However, the current oscillator implementations also have some shortcomings and problems. For example, the operational costs of a quantum annealer based on qubits are high and become complex due to the requirement of a low-temperature environment. In addition, although the optical coherent Ising computer has a competitive advantage over the quantum annealing machine, it needs a long fiber ring cavity for implementing Ising spin through time multiplexing. Additionally, it relies on an ultra-high-speed and huge-power-consumption field-programmable gate array (FPGA) to facilitate coupling in the measurement feedback scheme. The functioning of a digital CMOS annealing machine depends on random numbers, which are generated to introduce randomness, but maintaining real randomness is still technically challenging and requires a lot of post-processing. In our research, we will study and implement a local active memristor oscillator-based computer. The computer is using the negative resistance of the local active memristor and a capacitor, together with an external resistor, to constitute the oscillator, so the oscillator network forms a continuous-time dynamic system. The bistable of the oscillator phase simulates the Ising spin. The optimization problem is mapped by carefully selecting the coupling matrix for the memristor oscillatory network. The dynamic evolution of this physical system has reached the energy minimization point, which represents the solution of the optimization problem. The dynamics of this physical system have been explored in a wide range of applications, including understanding neural activities, realizing robot motion control, and using discrete-time Hopfield networks to solve optimization problems. Our work will be published in a series of papers.

In order to study the coupling in a local active memristor-based computer, the key is to study the interaction between oscillators. It is not practical to use the circuit simulation software SPICE’s (LTspice 24.0.9) transient simulation method because, with the expansion of the network scale, simulation time also increases significantly. This is because the single oscillator satisfies M individual differential equations; thus, the oscillator coupling network will have M×N differential equations to solve. In the past 40 years, the research has shifted to using the oscillator phase model to study the dynamic characteristics of the network, which will reduce the number of differential equations to N, saving a lot of simulation time. Among them, the most representative is the Kuramoto network model [41,42,43,44,45,46]. The advantage of this model is that the equation is simple, and it is easy to predict the phase evolution of the system. However, this model assumes that the oscillators are sinusoidally coupled, which limits its scope of application and accuracy. It is obvious that the network with local active memristor oscillators lacks sinusoidal coupling, so it is necessary to find another way to model them. In this paper, we use the research methods of neuro-dynamics for reference and introduce the concept of weak coupling [47,48]. We then solve the adjoint of the differential equations to further calculate the interaction function between the oscillators, expanding it into a Fourier series, so as to obtain the phase model of the system. Although our research object is the oscillator pair, the same research method can be easily extended to the oscillator coupling network, so as to provide a reference for the research on the memristor oscillatory network computer.

The structure of this paper is organized as follows: In Section 2, we briefly introduce the Kuramoto model because of its strong relevance to our research. We begin by detailing Winfree’s method, then Kuramoto’s, and then we proceed by comparing the two. In Section 3, we describe the analysis of the memristor phase dynamics comprehensively. Firstly, we give Chua’s unfolding model of the memristor, which is different from the traditional physical model. However, it has characteristics such as a simple structure, clear physical image, and ease of analysis. Secondly, as an additional segment, we will introduce the weak coupling theory because it is fundamentally important to our analysis. Thirdly, based on the theory and model, we solve the adjoint of the memristor differential equations so as to determine the interaction function between the oscillators. Subsequently, we analyze the case of one memristor oscillator. Fourthly, the interaction function of oscillator pairs is expanded into Fourier series, and then the phase model is obtained. Section 4 is the summary of our research.

## 2. The Kuramoto Model

The Kuramoto Model is a classical model of oscillator network. We will briefly introduce the model because of its strong connection to ours.

Suppose there are N interacting self-sustained limit-cycle oscillators in a network. The first assumption we make is that they are nearly identical. We expect the intrinsic frequencies of the oscillators to be sufficiently similar, so that they are represented within the same probability distribution, in which we have expected (mean) frequencies. We also know that the oscillators are affected by one another; they are “coupled” through metabolic, bioluminescent, electrical, mechanical, or other channels. In the absence of this coupling, each oscillator is left to operate independently and, therefore, maintains its own frequency, allowing no synchronization to occur. In modelling the evolution of individual oscillators within such a system, we start with this intrinsic frequency $\omega_{i}$, and then take into account the influence of all the other oscillators. We avoid further complication by ignoring outside interference, that is, anything outside of the network of oscillators, which may also influence their frequencies. This forms an ODE in an n-dimensional vector space.

Now that we have ensured the oscillators traverse the same limit cycle regardless of perturbation, albeit at differing frequencies, we have reduced the problem by shifting our focus away from the consideration of dynamics in an n-dimensional vector field to describing the pertinent inter-relations among functions of a single scalar variable, namely, the phase in the cycle. This just means that, as they traverse the same cycle, we will always know their position, simply by observing a single parameter: the phase.

Winfree derived his model as follows:(1)$$\frac{d\theta_{i}}{dt}=\omega_{i}+\left(\sum_{j\neq i}X(\theta_{j})\right)Z(\theta_{i}),\;\;\;\;\;i,j=1,\cdots N$$
where $\theta_{i\;(j)}$ is the phase. Winfree defined the concept of an “influence function” X, which determines the level of influence on an oscillator depending on its phase. He also defined the concept of a “sensitivity function” *Z*, which represents the change in frequency of an oscillator when perturbed by an arbitrarily small stimulus. It is also dependent on phase, so this tells us that the sensitivity of an oscillator depends on where it is within its oscillation period.

Whilst Winfree demonstrated synchronization numerically, his model was analytically difficult to solve. Kuramoto, taking the baton from Winfree’s approach, developed a solvable model in 1975. To model the phase evolution of the network, Kuramoto was primarily motivated by a desire to model synchronization behavior within chemical reaction–diffusion systems, though he reflected on the broader applicability of his model to other synergetic natural processes. He derived a phase model as follows:(2)$$\frac{d\theta_{i}}{dt}=\omega_{i}+\frac{V}{N}\sum_{j\neq i}\sin(\theta_{j}-\theta_{i}),\;\;\;\;\;i,j=1,\cdots N$$
where $\frac{V}{N}$ is the coupling strength. Comparing with (1), we can see that $\frac{V}{N}$ corresponds to *X* and *Z*, which take the form of $\sin(\theta_{j}-\theta_{i})$.

In the next section of this paper, we will see that *Z* does not necessarily take the form $\sin(\theta_{j}-\theta_{i})$. Instead, it will take a form depending on the nature of the oscillator and can be expanded as a Fourier series.

## 3. The Dynamical Characteristics of the Memristor

### 3.1. Chua’s Unfolding Model of the Memristor

According to the literature [49], the physical memristor of Figure 1a can be the equivalent to the structure in Figure 1b. An intrinsic memristor $\tilde{M}$ is connected in parallel with a nonlinear resistor R and then it is connected in series with the contact resistor $R_{C}$, where the intrinsic memristor’s equation is as follows:(3)$$\frac{dT}{dt}=g(T,{\tilde{v}}_{m})≜\frac{1}{C_{th}}\cdot {\tilde{i}}_{m}\cdot {\tilde{v}}_{m}-\frac{\Gamma_{th}}{C_{th}}\cdot (T-T_{amb}),$$
(4)$${\tilde{i}}_{m}=G(T,{\tilde{v}}_{m})≜\frac{1}{R_{01}}\cdot \exp(-\frac{a_{01}-a_{11}\cdot \left|{\tilde{v}}_{m}\right|}{T})\cdot {\tilde{v}}_{m},$$
where $C_{th}$ is the equivalent heat capacity, and $\Gamma_{th}$ is the equivalent thermal conductivity of the intrinsic Memristor $\tilde{M}$, respectively; $T_{amb}$ is the ambient temperature. $R_{01}$, $a_{01}$, $a_{11}$ are the fitting constants, $v_{m}$ and $i_{m}$ are Memristor’s voltage and current, respectively.

On the other hand, the constitutive equation of nonlinear resistance is as follows:(5)$$v_{R}=R_{02}\cdot i_{R}\cdot \exp(\frac{a_{02}-a_{12}\cdot \sqrt{\left|v_{R}\right|}}{T_{amb}}).$$
where $R_{02}$, $a_{02}$, $a_{12}$ are the fitting parameters, $v_{R}$ and $i_{R}$ are the voltage across the ends of the nonlinear resistor and the current flowing through, respectively. According to Equations (3)–(5), the equation of the memristor in Figure 1a can be derived by using Kirchhoff’s law of voltage and current. The corresponding differential algebraic equations is as follows:(6)$$\left\{\begin{array}{l} \frac{dT}{dt}=g(T,{\tilde{v}}_{m})≜\frac{1}{C_{th}}\cdot {\tilde{i}}_{m}\cdot {\tilde{v}}_{m}-\frac{\Gamma_{th}}{C_{th}}\cdot (T-T_{amb}),\\ {\tilde{i}}_{m}=G(T,{\tilde{v}}_{m})≜\frac{1}{R_{01}}\cdot \exp(-\frac{a_{01}-a_{11}\cdot \left|{\tilde{v}}_{m}\right|}{T})\cdot {\tilde{v}}_{m},\\ v_{R}=R_{02}\cdot i_{R}\cdot \exp(\frac{a_{02}-a_{12}\cdot \sqrt{\left|v_{R}\right|}}{T_{amb}}),\\ {\tilde{v}}_{m}=v_{R},\\ v_{m}=v_{R}+R_{C}\cdot ({\tilde{i}}_{m}+i_{R}). \end{array}\right.$$

It can be seen from observation that although this equation is based on physics, it is complex and difficult to reflect the physical meaning of the memristor, so we find another way to obtain the mathematical equation of the local active memristor by using Chua’s unfolding principle [50]. The empirical equations are as follows:(7)$$\dot{x}=g(x,v)≜a_{0}+a_{1}\cdot x+b_{2}\cdot v^{2}+c_{21}\cdot v^{2}\cdot x+c_{22}\cdot v^{2}\cdot x^{2}+c_{23}\cdot v^{2}\cdot x^{3}+c_{24}\cdot v^{2}\cdot x^{4}+c_{25}\cdot v^{2}\cdot x^{5},$$
(8)$$i=\hat{i}(x,v)≜G(x)\cdot v=(d_{0}+d_{1}\cdot x+d_{2}\cdot x^{2}+d_{3}\cdot x^{3}+d_{4}\cdot x^{4})\cdot v.$$
where i, *v*, and x represent device current, voltage, and state variables, respectively. According to the literature [51], the values of each fitting parameter in the formula are listed as follows (Table 1):

### 3.2. The Weak Coupling Theory

Weakly coupled oscillator theory can be used to quantify any form of coupling in an oscillator network’s phase-lock phenomena. As the name implies, the coupling between oscillators must be weak enough to make the quantization accurate. This means that in any given period, this coupling can only have a small impact on the dynamics of the oscillator. However, these minor effects may accumulate over many cycles and cause oscillators’ phase lock. The weak coupling theory allows the dynamics of each oscillator (which may have a high dimension) to be simplified to a single differential equation describing the phase of the oscillator. The autonomous differential equation considering the dynamics of the oscillator is:(9)$$\dot{X}=F(X),$$
where X is an n-dimensional vector, assuming that the equation has an asymptotically stable periodic solution $X_{0}(t)=X_{0}(t+T)$, where T is the period. Now couple the two oscillators defined by this equation, assuming that the variables are $X_{1}$ and $X_{2}$:(10)$${\dot{X}}_{1}=F(X_{1})+\varepsilon G_{1}(X_{2},X_{1}),$$
(11)$${\dot{X}}_{2}=F(X_{2})+\varepsilon G_{2}(X_{1},X_{2}),$$
where $G_{1}$ and $G_{2}$ are coupling functions, and ε is a small positive number, indicating the coupling strength. The weak coupling theory assumes that when the coupling is small enough, Equations (10) and (11) can be simplified to the phase equations
(12)$$X_{j}=X_{0}(\theta_{j})+O\left(\varepsilon \right),$$
(13)$${\dot{\theta }}_{1}=1+\varepsilon H_{1}(\theta_{2}-\theta_{1}),\;\;{\dot{\theta }}_{2}=1+\varepsilon H_{2}(\theta_{1}-\theta_{2}),$$
where $H_{j}$ is the function with period T. This model is called the phase model. A key question is how to calculate it. The calculation formula is as follows:(14)$$H_{j}(\varphi )=\frac{1}{T}\int_{0}^{T}X^{*}(t)\cdot G_{j}[X_{0}(t+\varphi ),X_{0}(t)]dt$$

This formula is a weighted average of $G_{j}$ with $X^{*}$ representing the coupling weight over a period T, where $X^{*}$ is a periodic function with period T, called the adjoint, which satisfies the linear differential equation and normalization conditions:(15)$${\dot{X}}^{*}(t)=-{\left[D_{X}F(X_{0}(t))\right]}^{T}\cdot X^{*}(t),\;\;X^{*}(t)\cdot {\dot{X}}_{0}(t)=1.$$
where $D_{X}F$ is the derivative matrix of F with respect to X, i.e., the Jacobian, denoted by ${\left[\;\right]}^{T}$, indicating the transpose of the matrix within brackets. Therefore, in order to calculate the function $H_{j}$, we must first calculate the adjoint solution of Equation (9) and the periodic integral (14), which can be calculated by numerical simulation software Matlab 2023b.

### 3.3. Single Memristor Oscillator

Figure 2 depicts a single memristor based oscillator circuit diagram, memristor M, first in parallel with the capacitor, and then in series with a bias resistor. The purpose of using the bias resistor is to bias the memristor in the local active region, that is, the negative resistance region, so as to form autonomous oscillation and show rich dynamic characteristics. The equations of the oscillator in Figure 2 are as follows:(16)$$\dot{x}=f_{1}(x,v)≜g(x,v),$$
(17)$$\dot{v}=f_{2}(x,v)≜\frac{1}{C}\cdot (\frac{V_{S}-v}{R_{S}}-\hat{i}(x,v)),$$
with the parameter list of each element in the figure being as follows (Table 2):

Figure 3 shows the voltage and current waveforms of the oscillator:

### 3.4. Phase Model of the Oscillator Pair

Next, we will analyze the coupling pair. Firstly, the resistance coupling is analyzed, and then the capacitance coupling is analyzed. The circuit of two oscillators coupled by resistance is shown in Figure 4:

The parameter values in the figure are shown in Table 2. Taking an appropriate value and changing within a certain range of $R_{C}$ will not affect the weak coupling dynamic characteristics of the circuit. The coupled pair in Figure 4 satisfies the following equations:(18)$${\dot{x}}_{1}=g(x_{1},v_{1}),$$
(19)$${\dot{v}}_{1}=\frac{1}{C_{1}}\cdot [\frac{V_{S1}-v_{1}}{R_{S1}}-{\hat{i}}_{1}(x_{1},v_{1})]+\frac{1}{C_{1}\cdot R_{C}}(v_{2}-v_{1}),$$
(20)$${\dot{x}}_{2}=g(x_{2},v_{2}),$$
(21)$${\dot{v}}_{2}=\frac{1}{C_{2}}\cdot [\frac{V_{S2}-v_{2}}{R_{S2}}-{\hat{i}}_{2}(x_{2},v_{2})]+\frac{1}{C_{2}\cdot R_{C}}(v_{1}-v_{2}),$$

Comparing Equations (19) and (21) with Equations (10) and (11), it can be seen that the coupling strength and coupling function are, respectively, listed as follows:(22)$$\varepsilon_{j}=\frac{1}{C_{j}\cdot R_{C}},\cdots j=1,2$$
(23)$$G_{1}=v_{2}-v_{1}$$
(24)$$G_{2}=v_{1}-v_{2}$$

Based on the above results, take the current as the object of investigation, and it is observed that the oscillator has a symmetrical structure. According to (14) and using Matlab, the interaction function H can be solved as shown in Figure 5:

This function can also undergo Fourier expansion to obtain the approximate expression:(25)$$\begin{array}{ll} H(\varphi )= & -8.023\times 10^{-8}+2.134\times 10^{-7}\cdot \cos\left(\varphi \right)-3.566\times 10^{-8}\cdot \cos(2\varphi )\\ & -1.235\times 10^{-9}\cdot \sin\left(\varphi \right)+4.126\times 10^{-10}\cdot \sin\left(2\varphi \right) \end{array}$$

Using the H function, we can get the phase model of the pair:(26)$$\begin{array}{l} {\dot{\theta }}_{1}=\omega_{1}+a\cdot H(\theta_{2}-\theta_{1}),\\ {\dot{\theta }}_{2}=\omega_{2}+a\cdot H(\theta_{1}-\theta_{2}), \end{array}$$
where $\theta_{1}$ and $\theta_{2}$ are the oscillator’s phases, respectively; $\omega_{1}=\omega_{2}$ is the oscillator frequency, which can be normalized to 1; a, the coupling coefficient, here can be taken as 0.1. At this point, we have analyzed the pair of resistance coupling. Next, we will analyze the case of capacitive coupling. In the context of a capacitive coupling pair, the circuit diagram is shown in Figure 6.

The parameter values in the figure are shown in Table 2. Taking an appropriate value and changing it within a certain range of $C_{C}$ will not affect the weak coupling dynamic characteristics of the circuit. The coupled pair illustrated in Figure 6 satisfies the following equations:(27)$${\dot{x}}_{1}=g(x_{1},v_{1}),$$
(28)$${\dot{v}}_{1}=\frac{1}{C_{1}}\cdot [\frac{V_{S1}-v_{1}}{R_{S1}}-{\hat{i}}_{1}(x_{1},v_{1})]-\frac{C_{C}}{C_{1}}\cdot \frac{1}{C_{1}+2C_{C}}\cdot \frac{1}{R_{S1}}[(v_{2}-v_{1})+R_{S1}\cdot ({\hat{i}}_{2}(x_{2},v_{2})-{\hat{i}}_{1}(x_{1},v_{1}))],$$
(29)$${\dot{x}}_{2}=g(x_{2},v_{2}),$$
(30)$${\dot{v}}_{2}=\frac{1}{C_{2}}\cdot [\frac{V_{S2}-v_{2}}{R_{S2}}-{\hat{i}}_{2}(x_{2},v_{2})]-\frac{C_{C}}{C_{2}}\cdot \frac{1}{C_{2}+2C_{C}}\cdot \frac{1}{R_{S2}}[(v_{1}-v_{2})+R_{S2}\cdot ({\hat{i}}_{1}(x_{1},v_{1})-{\hat{i}}_{2}(x_{2},v_{2}))],$$

Comparing Equations (28) and (30) with Equations (10) and (11), it can be seen that the coupling strength and coupling function are, respectively, as follows:(31)$$\varepsilon_{j}=\frac{C_{C}}{C_{j}}\cdot \frac{1}{C_{j}+2C_{C}}\cdot \frac{1}{R_{S_{j}}},\cdots j=1,2$$
(32)$$G_{1}=-[(v_{2}-v_{1})+R_{S1}\cdot ({\hat{i}}_{2}(x_{2},v_{2})-{\hat{i}}_{1}(x_{1},v_{1}))]$$
(33)$$G_{2}=-[(v_{1}-v_{2})+R_{S2}\cdot ({\hat{i}}_{1}(x_{1},v_{1})-{\hat{i}}_{2}(x_{2},v_{2}))]$$

Based on the above results, taking the current as the object of investigation, it is observed that the oscillator has a symmetrical structure. According to (14) and using Matlab, the interaction function H can be solved as shown in Figure 7.

This function can also undergo a Fourier expansion to obtain the approximate expression:(34)$$\begin{array}{ll} H(\varphi )= & -1.205\times 10^{-5}+4.648\times 10^{-6}\cdot \cos\left(\varphi \right)+4.640\times 10^{-6}\cdot \cos(2\varphi )\\ & -2.689\times 10^{-8}\cdot \sin\left(\varphi \right)-5.369\times 10^{-8}\cdot \sin\left(2\varphi \right) \end{array}$$

## 4. Summary

Based on Chua’s unfolding principle, this paper first provides the simple mathematical model of the Nb0x local active memristor. Based on this model, the dynamics of a single oscillator are analyzed. This oscillator uses the bias resistance to establish a bias for the memristor in the local active region or negative resistance region, thus forming autonomous oscillation. The voltage waveform and current flow through the device are given in this paper, respectively. On the basis of the above analysis, we take the memristor current as the research object. By solving the differential equation’s adjoint under the conditions of resistance coupling and capacitance coupling, the waveforms of the interaction function are obtained, respectively. Then, the Fourier expansion of this function is given. It is worth noting that although we only analyze the coupling of two oscillators, the same research method can be easily extended to an oscillation network containing multiple oscillators. This is of great significance for the study of the Ising machine based on the oscillator and for solving combinatorial optimization problems, including the maximum cut-set. If the number of original variables of the oscillator is M, the N oscillator coupling network becomes M×N. Using the transient simulation method in such cases consumes a lot of simulation time. However, using the phase model method, the free variable of the oscillator is reduced to the phase variable, resulting in the whole network having only N variables, which greatly accelerates the simulation time. When the number of oscillators is large, the advantage of this method is very obvious. In the subsequent phase, the phase model method will be used to analyze the dynamics of large-scale local active memristor oscillatory networks. This analysis aims to understand and construct a new computer based on memristors to solve NP-hard problems.

## Figures and Tables

**Figure 1 micromachines-15-00390-f001:**
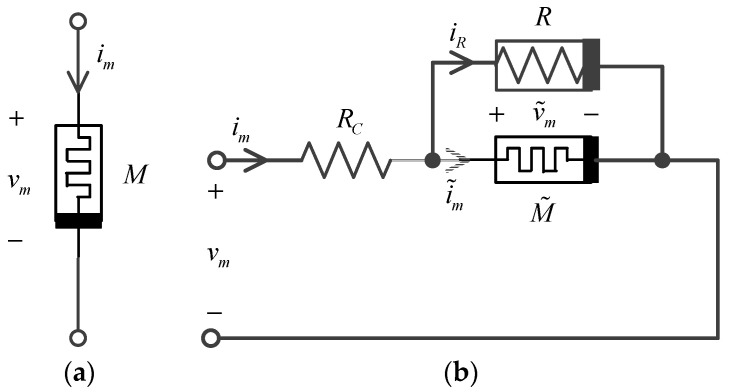
Nbox physical model of memristor. (**a**) The sign of the memristor; (**b**) The equivalent circuit of the memristor.

**Figure 2 micromachines-15-00390-f002:**
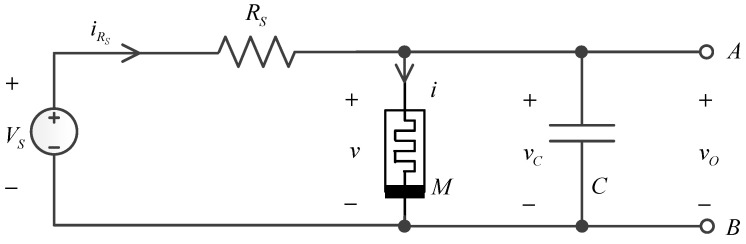
The single oscillator.

**Figure 3 micromachines-15-00390-f003:**
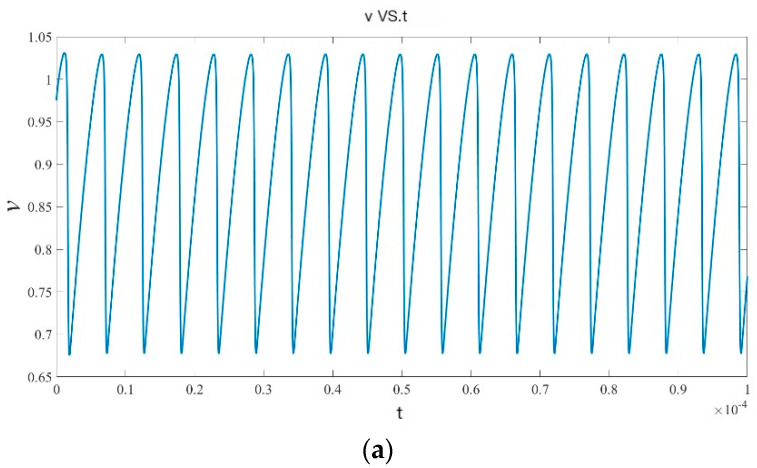
(**a**,**b**) first figure is the voltage waveform, and the second is the current waveform.

**Figure 4 micromachines-15-00390-f004:**
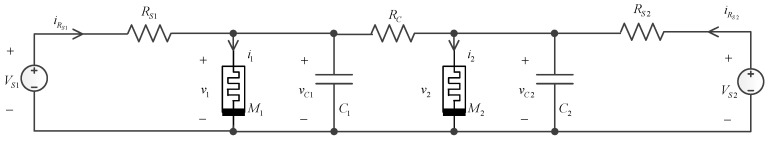
Memristor oscillator pair via resistance coupling.

**Figure 5 micromachines-15-00390-f005:**
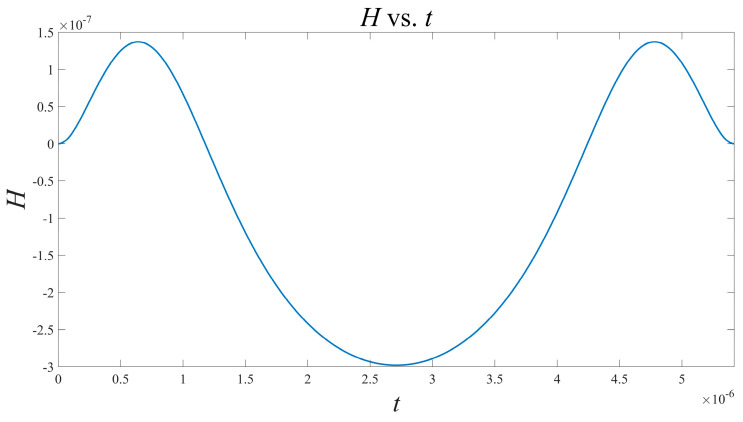
Waveform of H function versus time t in the presence of resistance coupling.

**Figure 6 micromachines-15-00390-f006:**
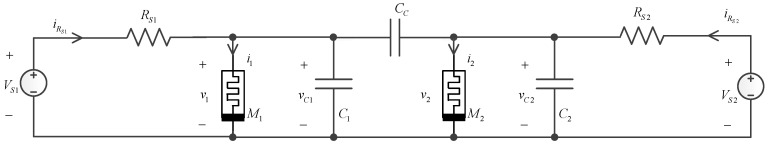
Oscillator pair via capacitive coupling.

**Figure 7 micromachines-15-00390-f007:**
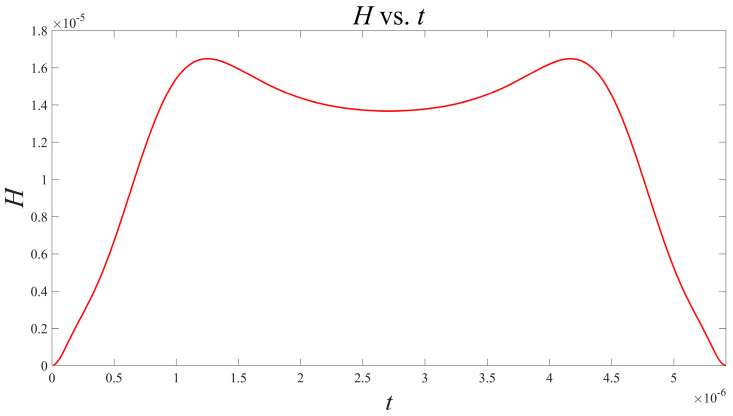
Waveform of H function versus time t during capacitive coupling.

**Table 1 micromachines-15-00390-t001:** Fitting parameters of the memristor Equations (7) and (8) [51].

$a_{0}$	$a_{1}$	$b_{2}$	$c_{21}$
$5.19\cdot 10^{9}$	$-2.05\cdot 10^{7}$	$7.21\cdot 10^{9}$	$-0.07\cdot 10^{9}$
$c_{22}$	$c_{23}$	$c_{24}$	$c_{25}$
$2.27\cdot 10^{5}$	$-2.4\cdot 10^{2}$	$1.25\cdot 10^{-1}$	$-2.69\cdot 10^{-5}$
$d_{0}$	$d_{1}$	$d_{2}$	$d_{3}$
$6.50\cdot 10^{-3}$	$-6.66\cdot 10^{-5}$	$2.14\cdot 10^{-7}$	$-2.14\cdot 10^{-10}$
$d_{4}$			
$1.19\cdot 10^{-13}$			

**Table 2 micromachines-15-00390-t002:** Element parameter values and initial conditions of differential equations [52].

$V_{S}$	$R_{S}$	C	Initial Condition
2.5 V	500 Ω	30 nF	(215,0.975 V)

## Data Availability

No new data were created.

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
