# Peer review of "A Phase Model of the Bio-Inspired NbOx Local Active Memristor under Weak Coupling Conditions"

_micromachines, 2024, doi:10.3390/mi15030390_

Round 1
Reviewer 1 Report
Comments and Suggestions for Authors
(1) The formatting of equations (3), (6), and (7)-(29) appears disorganized and inconsistent. I recommend unifying and centering the formatting of these equations to enhance the document's neatness and consistency..
(2) Figure 1, especially panels (a) and (b), along with Figures 4 and 6, have too much white space and are not centered, causing inconsistency in the document's figure formatting. For better visual coherence and professionalism, I suggest centering these figures and optimizing their spacing.
(3) The manuscript is missing a keywords section, which is essential for searchability and indexing purposes. I recommend adding a list of relevant keywords to improve the manuscript's visibility and adherence to academic publishing standards.
(4) Some paragraphs have issues with the first line indentation format.
(5) The first two headings starting from the Introduction are not numbered, contrary to the rest. Please ensure uniform numbering for consistency in formatting.
(6) The second and third main titles are too lengthy and lack conciseness. Consider incorporating these titles as subheadings under a single broader heading to enhance the clarity of the article's structure.
Comments on the Quality of English Languagecan be improved
Reviewer 2 Report
Comments and Suggestions for Authors
The paper “A phase model of the bio-inspired NbOx Local active memristor under weak coupling condition” describes first attempt to build a scalable computational model for the analysis of memristor networks.
The application of memristors may lead to new generation of nonvolatile resistive memory, neuromorphic processors and Ising computers.
However, after reading the paper, it is still unclear for me how the presented model may simplify the design or understanding of the above-mentioned circuits. The authors should modify the paper so that it would answer the following questions:
1. What is new compared to previous papers?
2. Is new model compatible with previous models, for instance, can it reproduce results of simpler models in special cases where they are accurate? Please provide the direct comparison for such case if possible.
3. Why is new model better? What exactly it does which was unaccessible before? Some examples should be given here.
4. Specific question about the interaction function H. Please describe its meaning and how knowing the function helps in designing, understanding or application of the circuit.
Minor points:
- Please simplify or rephrase the following sentence in the Introduction: “Memristor logic circuits mainly include memristor/CMOS hybrid logic circuits, memristor logic circuits, logic operation circuits based on programmable nanowire technology, embedded logic circuits, etc. [10,11].”
- The word “realize” at line 90 probably should be changed to “implement”
- Please use capital letter here: “Ising computer” (line 89)
- Kuramoto network model [41-26]: should be changed to [41-46]
- In the last paragraph of the introduction the authors start to describe the structure of the paper, but mention only the first part (“firstly, we give the Chua's unfolding model of the memristor”). Second, third etc. parts should be also mentioned.
- Please simplify the following sentence in the conclusion: “. On the basis of the above analysis, take the memristor current as the research object, analysis the dynamics of the oscillator pair, solving the differential equation’s adjoint under the condition of resistance coupling and capacitance coupling, the waveforms of the interaction function of the oscillator pair are obtained respectively”.
Comments on the Quality of English LanguagePlease simplify or rephrase the following sentence in the Introduction: “Memristor logic circuits mainly include memristor/CMOS hybrid logic circuits, memristor logic circuits, logic operation circuits based on programmable nanowire technology, embedded logic circuits, etc. [10,11].”
- The word “realize” at line 90 probably should be changed to “implement”
- Please use capital letter here: “Ising computer” (line 89)
- Kuramoto network model [41-26]: should be changed to [41-46]
- In the last paragraph of the introduction the authors start to describe the structure of the paper, but mention only the first part (“firstly, we give the Chua's unfolding model of the memristor”). Second, third etc. parts should be also mentioned.
- Please simplify the following sentence in the conclusion: “. On the basis of the above analysis, take the memristor current as the research object, analysis the dynamics of the oscillator pair, solving the differential equation’s adjoint under the condition of resistance coupling and capacitance coupling, the waveforms of the interaction function of the oscillator pair are obtained respectively”.
Round 2
Reviewer 1 Report
Comments and Suggestions for Authors
I endorse its publication
Reviewer 2 Report
Comments and Suggestions for Authors
OK, the authors have made some corrections
Comments on the Quality of English LanguageSome issues are still here